# ROYAL SOCIETY
# OPEN SCIENCE

analytical chemistry/statistics

calibration curve, Beer–Lambert Law, spectrometry, linear regression, concentration, absorbance

**Author for correspondence:**
Rosario Delgado
e-mail: delgado@mat.uab.cat

This article has been edited by the Royal Society of Chemistry, including the commissioning, peer review process and editorial aspects up to the point of acceptance.

# Misuse of Beer–Lambert Law and other calibration curves

## Rosario Delgado

Department of Mathematics, Universitat Autònoma de Barcelona, Campus de la UAB, 08193 Cerdanyola del Vallès, Spain

RD, 0000-0003-1208-9236

Calibration curves allow instrument calibration by predicting the concentration of an analyte in a sample from the reading of the instrument. This curve is constructed as the regression straight line that best fits the relationship between some known concentration standards and their respective instrument readings. An example is the Beer–Lambert Law, used to predict the concentration of a new sample from its absorbance obtained by spectrometry. The issue is that usually this methodology is misapplied. In this paper, we want to clarify this point, explaining what the error consists of and how (easily) to fix it, with the intention of ensuring that it does not continue to be reproduced in the experimental scientific work.

## 1. Introduction

Instrument calibration involves the construction of a calibration curve that allows to predict the concentration of an analyte in a sample from the reading of an instrument. This curve is the linear regression model that 'best fits' the relationship between some known concentration standards and the respective instrument responses. Of course, the effectiveness of the calibration procedure will depend on whether the relationship between the concentration and the instrument reading is indeed (approximately) linear. If it is, bivariate regression may be used to address the issue of predicting the output or dependent variable, say $Y$, from the input, regressor or independent variable $X$, by fitting a straight line to a scatterplot of observations on both variables, with the values of the variable $X$ on the $x$-axis (abscissa), and those of the variable $Y$ on the $y$-axis (ordinate). The best straight line, in the sense of minimizing the sum of the squared errors of prediction has the expression

$$y = b_0 + b_1 x,$$

$b_0$ being the intercept (where the straight line intersects the $y$-axis), and $b_1$ the slope, both computed from the observations (see formula (A 1) in appendix A), if the prediction for variable $Y$ when $X = x_0$ is that given by the straight line, that is, $b_0 + b_1 x_0$.

A paradigmatic example is the very popular Beer–Lambert Law (also known as Beer's Law), which establishes that under ideal conditions, the absorbance of a solution of an absorbing substance that is obtained by spectrometry techniques is directly proportional to the substance's concentration. This implies that the increase of the concentration value gives an increasing value of the absorbance, which is due to the fact that a high concentration of solution absorbs more light compared with a low concentration and that this happens in a linear way. This relationship between absorbance and concentration is used not only by chemists, but by experimental scientists of many other disciplines. Details of what this law says are given in §2.

There are innumerable works that collect research in which Beer's Law has been applied in very diverse fields that use the technique of spectrometry. Just to mention a few of them: in [1] the authors obtain the absorbance of some samples of glucose extracted from three different types of fruits peel wastes using UV–Vis spectroscopy, and from it and by means of Beer's Law, they obtain the corresponding concentrations, comparing between them. In [2] the authors say verbatim that 'The significance of Beer–Lambert Law is to measure the absorbance of a particular sample and to infer the concentration of the solution'. They use a spectrometer for measuring the absorbance of three macronutrients that are essential for plant growth (nitrogen, phosphorus and potassium) and are commonly used in fertilizers, in non-agriculture soil. As the quantity of fertilizer has to be estimated based on the requirements for optimum production, they apply the Beer–Lambert Law to determine the nutrients concentrations. Paper [3] explains a study for the determination of the amount of manganese metal present in tricalcium phosphate using flame atomic absorption spectrophotometer to observe the corresponding absorbance, by means of the calibration curve. The authors of [4] carry out an experiment to introduce a method to estimate the amlopidine in pure drug and marketed tablet Formulation consisting in the use of a calibration curve derived from Beer's Law to obtain the concentration from the absorbance. Andriamahenina *et al*. [5] investigate the effect of the presence of outliers in the calibration of lead by graphite furnace atomic absorption spectrometry, concluding that the presence of outliers worsens the quality of the measurement of the concentration of lead obtained from the absorbance given by the instrument reading, by using the calibration curve. A non-invasive alternative of blood glucose monitoring is introduced in [6], based on the detection of the optical density of the solution samples by means of a spectrophotometer, and then converting it into the corresponding glucose concentration by using the Beer–Lambert Law, with the help of a concentration curve. In [7] Ocean Optics Ocean View spectrometer operating software is used to obtain and process data from spectrometer, and get the transmittance (then, the absorbance) of a uric acid solution, from which to calculate uric acid concentration by using a concentration curve. The authors of [8] present and validate a quick and sensitive spectrophotometric method for quantitative determination of gliquidone in bulk drug, pharmaceutical formulations and human serum, based on the absorbance readings and their transformation into concentration through a calibration curve of the absorbance over the concentration. Restrepo *et al*. [9] report an easy methodology to construct handmade solar cells to produce clean energy from chlorophyll-a (chl-a) extracted from the leaves of Diacol Capiro potato. A spectroscopic calibration curve was constructed using different chl-a standard solutions and their absorbances. In [10] a quality-by-design (QbD) approach was implemented for the routine quality control analysis of serotonin in pharmaceutical dosage form through a spectroscopic method, by using a calibration curve of the absorbance over the concentration.

Although very common, Beer's Law is not the only source of application of calibration curves in different fields. For example, in the very recent paper [11] the authors construct calibration curves for the total protein eluted from membranes with respect to the concentrations of Bevacizumab or Trastuzumab used to add to serum employed to load the membranes. The total protein eluted from membranes is determined by measuring native fluorescence and then the concentration of Bevacizumab or Tratuzumab is determined using the calibration curve.

The problem of the proper use of calibration curves is common to many engineering and science applications, but not much attention has been paid to it from Statistics, with some exceptions (see ch. 15 in [12], for example, and references therein). The objective of this work is to show simply and without too many technicalities, in an accessible way to engineers and experimental scientists, the misuse of the calibration curves, explaining how to (easily) correct this pitfall, that could result in undesirable consequences. This issue has been treated before, although not always with the same success (see details in §4), but it is still worth reporting and publicizing to ward off further spreading among experimental scientists. Probably, in most cases this error has not practical importance and does not invalidate the published studies, since there will be little difference between the results obtained using the wrong calibration curve (classical calibration), and those obtained using the proper

one (inverse regression). Nevertheless, this does not prevent the error from being worth noting, for three main reasons:

(a) because regardless of the practical implications, from a conceptual point of view, the statistical methodology must be used in the appropriate way;
(b) because *a priori* it is not possible to know the extent of the repercussions of the misuse of the calibration curve on the results of an experiment;
(c) because an error does not cease to be so even though it is very generalized and commonly accepted.

The organization of the rest of the paper is as follows: in §3 we explain the misuse of the Beer–Lambert Law and other calibration curves. Section 4 details how to fix this problem, and a toy example of calibration is developed in §5 to show how the two calibration curves are applied, and compare them. Section 6 includes a few words in conclusion and an outline of what calibration curve is appropriate in every situation in figure 6. Finally, in appendix A we recall the main formulae of the linear regression model, and in appendix B we show two more examples of calibration, one with real experimental data and the other using simulation.

## 2. The Beer–Lambert Law

A spectrophotometer is an instrument that measures the number of photons delivered by a solution of a chemical species that absorbs light of a particular wavelength in a given unit of time, which is called the intensity, allowing to compare the intensity of the beam of light entering the solution ($I_0$) with the intensity of the beam of light exiting it ($I$). The ratio of these intensities is called transmittance, and is denoted by the letter $T$. That is, $T = I/I_0$. If the transmittance is a measure of the quantity of photons passing through a solution (the proportion of the intensity of the light entering the solution that exits), the absorbance $A$ is a measure of how much light is absorbed by the solution, and is defined as a function of the transmittance in this way,

$$A = -\log_{10}(T), \tag{2.1}$$

(large values of absorbance are associated with very little light passing through the solution, and on the opposite, small values of absorbance are associated with most of the light passing entirely through it).

When passing a beam of light of the appropriate wavelength through the solution, if it is fairly dilute, the photons will encounter a small number of the absorbing chemical species and then we can expect a high transmittance and low absorbance. On the contrary, if the solution is highly concentrated we will expect a higher number of the absorbing chemical species and a low transmittance and high absorbance. This leads us to think that the absorbance could be a monotonic increasing function of the concentration of the solution, and even that it could be (directly) proportional to it. As well, it seems that the absorbance would increase if the beam of light goes through the solution for a longer period of time, and since the speed of light is constant, we could think that the absorbance is also directly proportional to the path length of the beam through the solution. In this way we come to the (deterministic) Beer–Lambert Law, which states the following:

$$\text{The Beer} - \text{Lambert Law}: \quad A = \varepsilon L c, \tag{2.2}$$

where $c$ is the concentration of the absorbing species in the solution, $L$ is the path length of beam through the sample compartment where the solution is, and $\varepsilon$ is the proportionality constant. If the path length $L$ is reported in centimetres (cm), and the concentration $c$ is reported in molarity (moles per litre, mol l$^{-1}$), the proportionality constant $\varepsilon$ is called the molar absorptivity or molar extinction coefficient, and has units litres per mole-centimetre (l (mol × cm)$^{-1}$). In this way, when multiplying $\varepsilon$, $L$ and $c$, all the units cancel and as such, it follows that absorbance $A$ is unit-less. Note that $\varepsilon$ is intrinsic to the absorption of the solution of chemical species at a particular wavelength of light.

If, in a given context, we know three of the four quantities that appear in equation (2.2), we can solve for the value of the fourth. We could obtain the absorbance of a solution $A$ from its concentration $c$, knowing the other two quantities $L$ and $\varepsilon$, without needing more to substitute in (2.2). Or vice versa, knowing the absorbance of the solution at a given wavelength, usually from the transmittance, by using equation (2.1), we could obtain the concentration by solving $c$ from equation (2.2),

$$c = \frac{A}{\varepsilon L}, \tag{2.3}$$

(note that equations (2.2) and (2.3) are completely equivalent, since $\varepsilon L > 0$).

The crux of the issue appears when the product of the molar absorptivity and the path length, $\varepsilon L$, which is constant for a given solution ($\varepsilon$) and as long as the same sample compartment is used to make measurements ($L$), is not known. Then, in order to determine the concentration $c$ of the solution given its absorbance value $A$, a calibration curve needs to be constructed. And it is at this point that the source of the error appears, as will be described in the next section.

## 3. Misuse of the calibration curves

What is this widespread error? In the context of lack of knowledge of the (constant) value of $\varepsilon L$, the following misuse of the Beer–Lambert Law is usually committed: in order to construct the calibration curve to predict the concentration of an unknown solution from its known absorbance, a set of standard concentrations within the range of the measuring instrument are prepared, and the corresponding absorbances are determined by spectrometry, say $(c_1, A_1)$, $(c_2, A_2)$, ..., $(c_n, A_n)$. Then the equation of the regression straight line for the response variable absorbance and prediction variable (or regressor) concentration that best fits these $n$ points is

$$\text{Calibration curve of } A \text{ over } c: \quad A = \beta_0 + \beta_1 c, \tag{3.1}$$

where $\beta_1$ is the slope of the line, and $\beta_0$ is the $y$-intercept, and both are obtained from the $n$ points by means of the linear least-squares method and are given by formulae

$$\beta_1 = \frac{\sum_{i=1}^n c_i A_i - n \bar{c} \bar{A}}{\sum_{i=1}^n c_i^2 - n (\bar{c})^2}, \quad \beta_0 = \bar{A} - \beta_1 \bar{c}, \quad \text{with } \bar{c} = \frac{1}{n} \sum_{i=1}^n c_i, \; \bar{A} = \frac{1}{n} \sum_{i=1}^n A_i. \tag{3.2}$$

Now, if we denote by $\widehat{A}_i$ the prediction of the absorbance given by the straight line for a solution whose concentration is that corresponding to the $i$th point, $c_i$, it is obtained by substituting $c_i$ into equation (3.1),

$$\widehat{A}_i = \beta_0 + \beta_1 c_i,$$

then the difference (error) between the predicted and the observed absorbance for the solution with concentration $c_i$ is: $e_i = A_i - \widehat{A}_i$, and formulae in (3.2) are obtained imposing that the sum of the square of the errors be minimum

$$\text{SSE} = \sum_{i=1}^n e_i^2 = \sum_{i=1}^n \left( A_i - \widehat{A}_i \right)^2 = \sum_{i=1}^n \left( A_i - (\beta_0 + \beta_1 c_i) \right)^2. \tag{3.3}$$

That is, if absorbance $A$ is plotted versus concentration $c$ for the series of $n$ known solutions with the dependent variable $A$ placed on the $y$-axis, and the independent variable $c$ graphed on the $x$-axis, the calibration curve (3.1) is the straight line that best fits the $n$ points in the plane in the sense of minimizing the sum of the squares of the distances from each point to its prediction vertically (figure 1).

Calibration curve (3.1) is therefore intended for predicting the absorbance of new solutions for which concentrations are known, since with the parameters $\beta_0$ and $\beta_1$ given by (3.2), it ensures that the sum of the square of the errors committed in prediction for the $n$ initial solutions is as low as possible. Then, given the concentration of a new solution, say $c_0$, we can obtain the predicted absorbance value for it, $\widehat{A}_0$ from equation (3.1) by substituting the concentration $c_0$, that is $\widehat{A}_0 = \beta_0 + \beta_1 c_0$ (figure 2a). However, in what is known as classical calibration, (3.1) is usually used inappropriately to predict the concentration of new solutions for which absorbances are known in the following way: first finding the $y$-value on the regression straight line corresponding to the measure of the absorbance, and then tracing downward to see which concentration matches up to it, and this value will be the predicted concentration of the solution with that absorbance (figure 2b).

That is, given the absorbance value of a new unknown solution, say $A_0$, the usual (wrong) practice is to obtain the predicted concentration value for it, $\widehat{c}_0$, from equation (3.1) by substituting the absorbance $A_0$, that is

$$\widehat{c}_0 = \frac{A_0 - \beta_0}{\beta_1} = \frac{-\beta_0 + A_0}{\beta_1} = b + m A_0, \tag{3.4}$$

where $b = -\beta_0/\beta_1$ and $m = 1/\beta_1$, being $\beta_0$ close to zero, and $\beta_1$ an estimation of the unknown product $\varepsilon L$, both computed using the formulae in (3.2). If we predict the concentration for the $i$th point given its

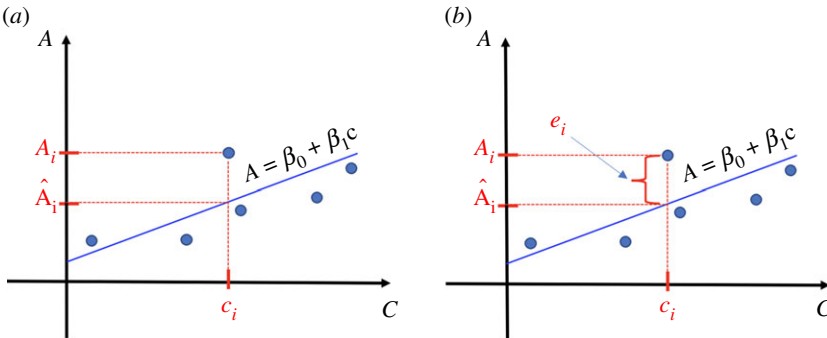

**Figure 1.** Calibration curve of $A$ over $c$ properly used to predict absorbance $A$ from concentration $c$. The error of prediction is $e_i = A_i - \widehat{A}_i$ (b).

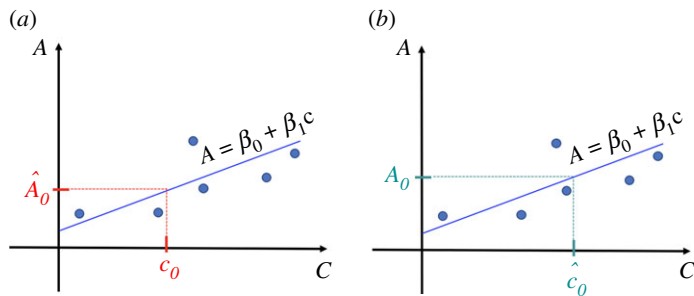

**Figure 2.** (a) Calibration curve of $A$ over $c$ properly used to predict absorbance from concentration and prediction of the absorbance $\widehat{A}_0$ of a new solution from its concentration $c_0$. (b) Calibration curve of $A$ over $c$ misused to predict concentration from absorbance (classical calibration) and prediction of the concentration $\widehat{c}_0$ of a new solution from its absorbance $A_0$.

absorbance in this (wrong) way, we obtain

$$\widehat{c}_i = \frac{A_i - \beta_0}{\beta_1}. \tag{3.5}$$

But then, the sum of squared errors (differences between observed and predicted concentrations) is

$$\sum_{i=1}^{n} \left( c_i - \widehat{c}_i \right)^2 = \sum_{i=1}^{n} \left( c_i - \frac{A_i - \beta_0}{\beta_1} \right)^2,$$

and we do not have any optimality result in the sense that we cannot ensure that it is as small as possible, with $\beta_0$ and $\beta_1$ given by (3.2), unlike what happens with (3.3).

In summary: it is possible algebraically to predict the concentration from the absorbance by using the calibration curve of the absorbance $A$ over the concentration $c$ given by (3.1), following the expression (3.4) with $\beta_0$ and $\beta_1$ given by (3.2), as in figure 2b. This is the classical calibration approach. But this is not the optimal way, since we do not control for the prediction errors that are committed. Therefore, this procedure should be avoided. Instead, it is advisable to preserve (3.1) exclusively to predict the absorbance from the concentration, because this procedure is optimal to achieve the minimum sum of the squared prediction errors (figure 2a).

## 4. Easily fixing it

The problem is easily solvable: since it is a question of constructing a calibration curve to predict the concentration of a new solution of which the absorbance is known, from the concentrations and absorbances of the initial known solutions, the regression straight line of the concentration $c$ over the absorbance $A$ will be the proper one to be used, since it is the one that minimizes the sum of the squared errors of prediction (ordinary least squares, OLS). From the known concentrations and absorbances of the set of $n$ solutions, we obtain the equation of the regression straight line for the

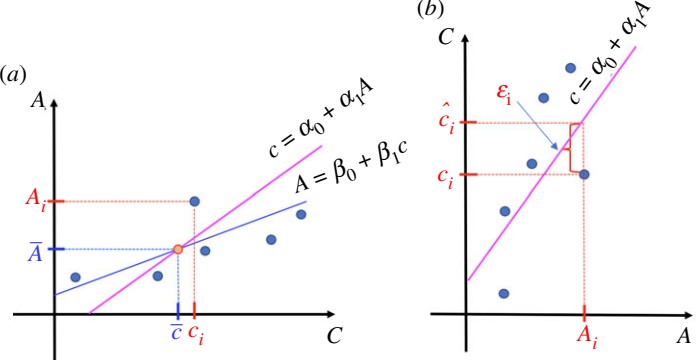

**Figure 3.** (a) Calibration curve of $A$ over $c$ (blue colour) and calibration curve of $c$ over $A$ (magenta colour), on the same coordinate axes. The intersection point of the two lines is $(\bar{c}, \bar{A})$. (b) Calibration curve of $c$ over $A$ interchanging the axes, with the prediction error $\varepsilon_i = c_i - \widehat{c_i}$.

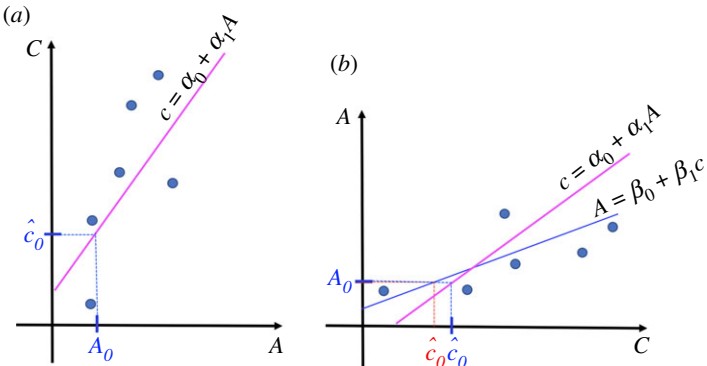

**Figure 4.** Predicting the concentration $\widehat{c_0}$ of a new solution from its absorbance $A_0$. (a) In blue, with the calibration curve of $c$ over $A$ (inverse regression). (b) Comparison with the prediction using the calibration curve of $A$ over $c$ (classical calibration) in red, on the same coordinate axes.

response variable concentration and prediction variable absorbance

$$\text{Calibration curve of } c \text{ over } A: \quad c = \alpha_0 + \alpha_1 A, \tag{4.1}$$

with the slope $\alpha_1$, which is an estimation of $(\varepsilon L)^{-1}$, and the intercept $\alpha_0$ (close to zero) obtained from the formulae

$$\alpha_1 = \frac{\sum_{i=1}^{n} c_i A_i - n\bar{c}\bar{A}}{\sum_{i=1}^{n} A_i^2 - n(\bar{A})^2}, \quad \alpha_0 = \bar{c} - \alpha_1 \bar{A}. \tag{4.2}$$

Given the absorbance corresponding to the $i$th point, $A_i$, the prediction of its concentration, $\widehat{c_i}$, is obtained by substituting $A_i$ into equation (4.1), that is,

$$\widehat{c_i} = \alpha_0 + \alpha_1 A_i, \tag{4.3}$$

and then the difference (error) between the predicted and the observed concentration for the solution with absorbance $A_i$ is: $\varepsilon_i = c_i - \widehat{c_i}$, and in these cases formulae in (4.2) are obtained imposing that the following sum of the square of the errors be minimum:

$$\sum_{i=1}^{n} \varepsilon_i^2 = \sum_{i=1}^{n} (c_i - \widehat{c_i})^2 = \sum_{i=1}^{n} (c_i - (\alpha_0 + \alpha_1 A_i))^2,$$

(see figure 3). Note that the two straight lines (4.1) and (3.1) intersect at the point $(\bar{c}, \bar{A})$.

Given the absorbance of a new solution, say $A_0$, we can obtain the predicted concentration value for it, $\widehat{c_0}$ from equation (4.1) by substituting the absorbance $A_0$ in this direct way

$$\widehat{c_0} = \alpha_0 + \alpha_1 A_0, \tag{4.4}$$

and if we compare (4.4) with (3.4) we realize that in general, $\alpha_0 \neq b$ and $\alpha_1 \neq m$, that is, the two approaches are not equivalent, as can be seen graphically in figure 4.

Since we are interested in minimizing the sum of the squared errors of prediction, it is then evident that the proper calibration curve is (4.1) and not (3.1). This approach is known as inverse regression from [13]. It is perfectly adequate in terms of prediction errors, since the OLS method does not depend on any additional hypotheses about the regression model, being the optimal approach in the sense of minimizing the sum of the squared errors.

However, it is true that to make statistical inferences about the linear regression model (confidence intervals and tests of hypothesis on the coefficients of the regression straight line), some hypotheses are assumed (see appendix A for details), being the most basic that the regressor is measured without error, and that the response variable is randomly distributed following a normal distribution with mean a linear function of the regressor, and constant variance. We will call them: LR hypotheses (by linear regression). If we are interested in making statistical inferences about the regression model, we have to design the experiment to collect data in such a way that these assumptions are reasonably fulfilled. In our case, this means that absorbances have to be measured with precision while concentrations are measured with non-negligible error, which in practice may not be possible, and this is considered in the literature the weak point of the inverse regression approach. Indeed, in the opinion of Parker *et al.* [14], for example, the observed measurements (absorbances) in practical calibrations are subject to measurement error, violating the LR hypotheses.

What if the LR hypotheses with regressor the absorbance and the concentration as response, corresponding to the approximation of the inverse regression, are not fulfilled, not even roughly? Nothing invalidatesthis approximation, in our opinion, for the following reasons:

(1) The hypotheses are needed if we want to make statistical inference about the model, not to make predictions, that can be carried out equally.
(2) The convenience of using the inverse regression approach relies on OLS, which does not depend on any hypothesis but on the errors of prediction, which allow to evaluate the predictive capacity of any model.
(3) The greater predictive power of the inverse regression, compared with that of classical calibration, gives support to its use and has been shown empirically in this work by a toy example in §5 and two more examples in appendix B, one with real experimental data, and the other built by simulation.

   Likewise, it has also been described in some works. In this regard, [13] compared classical calibration (named there Method A) and inverse regression (Method B) using simulations, and recommended the latter based on the mean squared error. The authors of [14] also arrived at the same conclusions through some simulation studies (see also references therein in the same vein), although other authors criticize that recommendation. For example, in the recent paper [15], the authors introduced a new methodology, the 'reverse inverse regression' to address the same problem, assuming that the inputs (concentration values) vary according to Gaussian distributions, which allow them to derive some statistical properties, and criticize the inverse regression approach based on the treatment of the inputs (absorbance values) as determined with small error. But they compare their method against classical calibration and inverse regression using a simulation study, and have to recognize the best behaviour of the latter in the sense of minimizing the variance of the prediction interval.

In brief, leaving aside assumptions that could, or not, be accomplished (that in case to be fulfilled allow to deduce some statistical properties for the linear regression model), if we are interested in prediction, the best approach nonetheless seems to be inverse regression.

## 5. A toy example

We prepare a set of $n$ ( = 10) standards within the range of the measuring instrument, with the following made-up values of concentration (in mg l$^{-1}$) and absorbance, recorded in table 1.

The two calibration curves given by (3.2) and (4.2) are:

Classical calibration (curve of $A$ over $c$): $A = \beta_0 + \beta_1 c = 0.00554 + 0.0003936364 c$

Inverse regression (curve of $c$ over $A$): $c = \alpha_0 + \alpha_1 A = 6.06475 + 2128.07645 A$

We can observe in figure 5 that indeed, as explained above, the two curves are not the same, and they cut at the point ($\bar{c} = 110$, $\bar{A} = 0.04884$). Moreover, the values of the $R$-squared ($R^2$) have also been reported for the two calibration curves, being higher than that of the inverse regression approach to

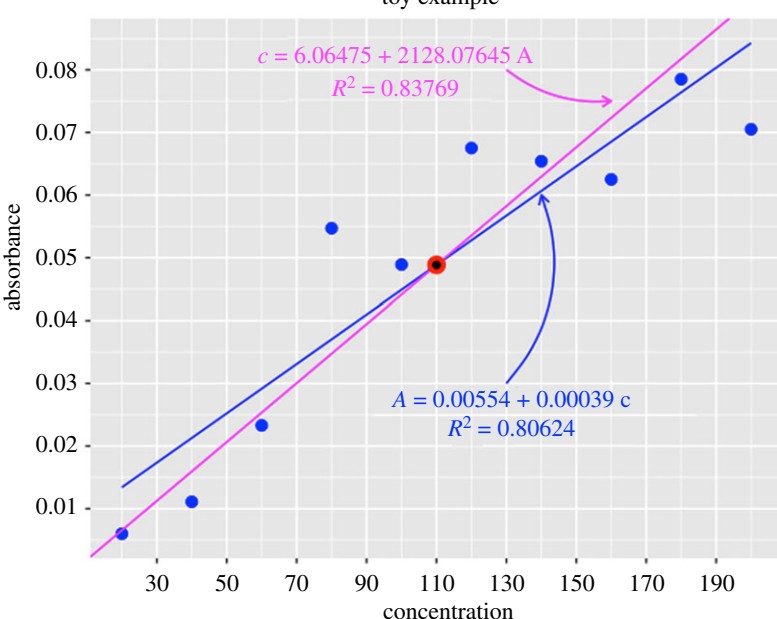

**Figure 5.** Calibration curves for the toy example to predict the concentration from the absorbance. In blue the calibration curve of $A$ over $c$ (classical calibration). In magenta the (proper) calibration curve of $c$ over $A$ (inverse regression).

**Table 1.** Toy example: concentration and absorbance of 10 solutions, and their averages.

| concentration (mg l$^{-1}$) | absorbance |
|---|---|
| 20 | 0.0060 |
| 40 | 0.0111 |
| 60 | 0.0233 |
| 80 | 0.0547 |
| 100 | 0.0489 |
| 120 | 0.0675 |
| 140 | 0.0654 |
| 160 | 0.0625 |
| 180 | 0.0785 |
| 200 | 0.0705 |
| $\bar{c} = 110$ | $\bar{A} = 0.04884$ |

predict concentration from absorbance. $R^2$ represents the proportion of variation in the response variable that is explained by the calibration curve (the higher the better).

Note that $R^2 = 1 - (\text{SSE}/\text{SST})$, where SSE and SST denote the sum of squared errors and the sum of squared total, respectively, that is, $\text{SSE} = \sum_{i=1}^{n}(c_i - \widehat{c}_i)^2$ and $\text{SST} = \sum_{i=1}^{n} c_i^2 - n\,(\bar{c})^2$, being $\widehat{c}_i$ the prediction for the concentration of the $i$th solution (with absorbance $A_i$), that is given by (3.5) for the classical calibration approach, but by (4.3) for the inverse regression. In table 2, we report the predictions $\widehat{c}_i$ with the two approaches.

As expected, the proper calibration curve (that of $c$ over $A$) has lower standard error (s.e.) and higher $R^2$ than the usual one (the calibration curve of $A$ over $c$), to predict concentration from absorbance, which confirms the theoretical result that states that it is better. In other words, inverse regression is better than classical calibration in the sense of minimizing the sum of squared errors in prediction, and this conclusion is independent of the hypotheses of the linear regression model.

One way to see if the differences in prediction errors are statistically significant is as follows: consider the differences of the absolute value of the prediction errors with the two approaches (last column in

**Table 2.** Toy example: predictions with the two methods: classical calibration and inverse regression, and corresponding prediction errors with the difference of the absolute value of the errors. In italics the maximum $R^2$ and the minimum standard error (s.e.), as well as the $p$-value for the one-sided $t$-test in favour of the hypothesis that the mean of the differences is greater than 0.

| $A_i$ | $c_i$ | predictions $\widehat{c_i}$ | | errors $c_i - \widehat{c_i}$ | | |
| | | classical $\frac{A_i - \beta_0}{\beta_1}$ | inverse $\alpha_0 + \alpha_1 A_i$ | classical $e_i$ | inverse $\varepsilon_i$ | difference $|e_i| - |\varepsilon_i|$ |
|---|---|---|---|---|---|---|
| 0.0060 | 20 | 1.168591 | 18.83320 | 18.83141 | 1.166795 | 17.66461345 |
| 0.0111 | 40 | 14.124711 | 29.68639 | 25.87529 | 10.313605 | 15.56168328 |
| 0.0233 | 60 | 45.117783 | 55.64893 | 14.88222 | 4.351073 | 10.53114443 |
| 0.0547 | 80 | 124.886836 | 122.47.053 | −44.88684 | −42.470528 | 2.41630800 |
| 0.0489 | 100 | 110.152425 | 110.12768 | −10.15242 | −10.127685 | 0.02474035 |
| 0.0675 | 120 | 157.404157 | 149.70991 | −37.40416 | −29.709907 | 7.69425040 |
| 0.0654 | 140 | 152.069284 | 145.24095 | −12.06928 | −5.240946 | 6.82833797 |
| 0.0625 | 160 | 144.702079 | 139.06952 | 15.29792 | 20.930476 | −5.63255415 |
| 0.0785 | 180 | 185.348730 | 173.11875 | −5.34873 | 6.881252 | −1.53252256 |
| 0.0705 | 200 | 165.025404 | 156.09414 | 34.97460 | 43.905864 | −8.93126815 |
| | | SSE = | | 6394.129 | 5356.287 | Shapiro–Wilk $p$-value =0.915 |
| | | MSE = SSE/$(n - 2)$ = | | 799.266 | 669.536 | one-sided $t$-test for mean >0 |
| | | s.e. = $\sqrt{\text{MSE}}$ = | | 28.271 | *25.875* | $p$-value = *0.07094*$^*$ |
| | | $R^2 = 1 - \text{SSE/SST}$ = | | 0.80624 | *0.83769* | |

$^*$Significance at 10% level.

**Table 3.** Radius of the (approximated) prediction intervals, and $p$-value of the one-sided $t$-test in favour of the hypothesis that the mean of the differences of the radius is greater than 0.

| $A_i$ | $c_i$ | prediction interval radius | | |
| | | classical ($a$) | inverse ($b$) | difference ($a$) − ($b$) |
|---|---|---|---|---|
| 0.0060 | 20 | 13.47537 | 12.76106 | 0.6607362 |
| 0.0111 | 40 | 13.28582 | 12.60487 | 0.6366435 |
| 0.0233 | 60 | 12.90608 | 12.29470 | 0.5882947 |
| 0.0547 | 80 | 12.57598 | 12.02834 | 0.5462853 |
| 0.0489 | 100 | 12.55686 | 12.01301 | 0.5438552 |
| 0.0675 | 120 | 12.74684 | 12.16581 | 0.5680196 |
| 0.0654 | 140 | 12.70719 | 12.13383 | 0.5629739 |
| 0.0625 | 160 | 12.65973 | 12.09561 | 0.5569352 |
| 0.0785 | 180 | 13.02142 | 12.38851 | 0.6029853 |
| 0.0705 | 200 | 12.81089 | 12.21756 | 0.5761737 |
| Shapiro–Wilk $p$-value = | | | | 0.1859 |
| one-sided $t$-test for mean >0 $p$-value = | | | | $1.998 \times 10^{-12}$*** |

***Significance at 0.1% level.

table 2). For this sample of size 10, we can perform a goodness-of-fit test for normality (Shapiro–Wilk test) obtaining a $p$-value of 0.915, which does not allow us to reject the hypothesis of normality, so we apply the one-sided $t$-test to compare the mean against 0, giving a $p$-value of 0.07094$^*$. This $p$-value is not less than 0.05 but it is not very far off (it is less than 0.10), so we can say that there is a slight statistical significance in favour of the difference of the absolute values of the predictive errors being positive, or what is the same, that on average the errors with the classical calibration approach are

greater in absolute value than with the inverse regression. Since in practical calibrations the errors in making the predictions are of the most important measures of the goodness of the calibration method, in table 3 we also record the values of the radius of the prediction intervals.

For any absorbance $A_i$, the corresponding prediction intervals are of the form $\widehat{c}_i \pm (a)$ using the classical calibration (the expression for $(a)$, which has been derived with the approximative Delta method, can be found in (A 6), appendix A), and $\widehat{c}_i \pm (b)$ with the inverse regression, where by (A 5) in appendix A, $(b) = t_{1-(\alpha/2)}^{n-2} \sqrt{\left(\sum_{i=1}^{n} \varepsilon_i^2/(n-2)\right)\left(1 + (1/n) + (A_i - \bar{A})^2/\left(\sum_{i=1}^{n} A_i^2 - n\,\bar{A}^2\right)\right)}$, with $\varepsilon_i = c_i - (\alpha_0 + \alpha_1 A_i)$.

Note that both $(a)$ and $(b)$ in table 3 are deduced from the assumptions of the linear regression model; therefore, they will be more or less adjusted, depending on the degree of compliance with the LR hypotheses. In any case, for all absorbance values, the estimated radius of the prediction interval is greater with the classical calibration than with the inverse regression. This fact is statistically significant: if the two methods were equivalent from the perspective of the prediction interval error, or if the classical calibration were better, the probability that for the 10 absorbance values the prediction interval radius with the inverse regression are all less than the corresponding with the classical calibration, is upper bounded by

$$P(B(10, \, p = 0.5) = 10) = 0.5^{10} = 0.0009765625^{***},$$

which is a very low $p$-value (corresponding to the exact binomial test). This means that the probability that the 10 prediction interval radius with the inverse regression are less than the corresponding with the classical calibration if the first method is not better than the second in the sense of having less prediction error, is very low, which reveals that the assumption must be rejected, and accepted that inverse regression is statistically significantly better than classical calibration. The same conclusion is reached by performing a statistical one-sided $t$-test to compare the mean of the differences $(a)$–$(b)$ with 0, with a $p$-value of $1.998 \times 10^{-12***}$ in favour that the mean is greater than 0 or, equivalently, that on average, the radius of the prediction intervals for the classical calibration are greater than for the inverse regression. The $t$-test is performed after using a Shapiro–Wilk test of normality, whose $p$-value is: 0.1859.

As a final comment in this toy example, note that the analysis of variance (ANOVA) methodology for regression (see appendix A) can only be applied to the inverse regression approach, and that in this case, the ANOVA table is:

| source of variation response $c$ | d.f. | sum Sq | mean Sq | F-value |
|---|---|---|---|---|
| regressor $A$ | 1 | $\alpha_1^2 S_{AA} = 27643.713$ | $\alpha_1^2 S_{AA} = 27643.713$ | $f = 41.28787$ |
| residuals (error) | 8 | $SSE = \sum_{i=1}^{n} e_i^2 = 5356.287$ | $MSE = 669.536$ | |
| total | 9 | $SST = \sum_{i=1}^{n}(c_i - \bar{c})^2 = 33\,000$ | | |

where $S_{AA} = \sum_{i=1}^{n}(A_i - \bar{A})^2 = 0.006104104$. Then, if the LR hypotheses hold, the null hypothesis $H_0$: 'no linear relationship between $A$ and $c$' is rejected since the corresponding $p$-value is $P(F_{1,8} > 41.28787) = 0.0002035^{***}$. That is, we accept with a very strong statistical significance that $A$ and $c$ are linearly related. We observe the concordance between values in this ANOVA table and that of table 2. However, this is not true with classical calibration, the other approach. The reason is clear: the values recorded in its ANOVA table (that we have not reproduced here) are that of the regression curve of $A$ over $c$: $A = \beta_0 + \beta_1 c$ when used to predict the absorbance from the concentrations, and not vice versa. For this reason, to compare both approaches, the ANOVA methodology does not turn out to be useful.

# 6. Conclusion

There are many very painstaking experimental works in which an analytical methodology to determine the concentration of a given substance by using spectrometry is described. Without trying to undermine the interest of these studies, it is necessary to mention that in them, in a systematic way, a gross error is made in the application of the Beer–Lambert Law that allows to determine the concentration $c$ from the absorbance $A$. The pitfall consists in using the calibration curve of $A$ over $c$ (classical calibration), which is clearly not an optimal approach (see [13], for example), instead of using the calibration curve of $c$ over $A$,

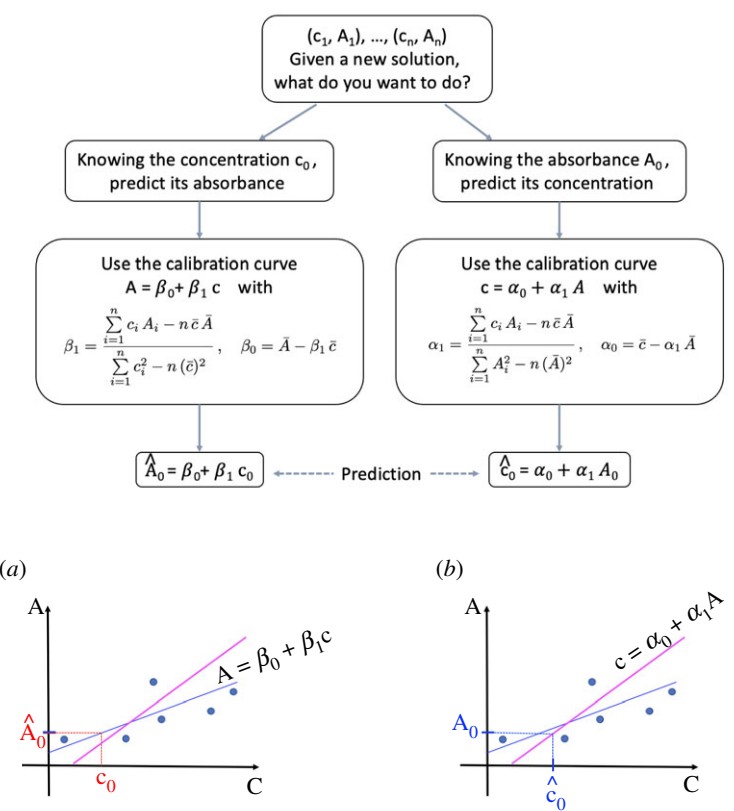

**Figure 6.** Outline on how to choose the most suitable calibration curve in each situation to get the proper prediction.

which would be the appropriate (inverse regression), in the sense of minimizing the sum of the squared errors of prediction.

But this not only happens in the application of Beer's Law: it is also a common practice in other contexts where instrument calibration is used, when inexpensive and quick measurements ($Y$) are related to expensive and time-consuming measurements ($X$) based on a set of observations, and we are interested in estimating the expensive measurement of $X$ given a new measurement of $Y$. Instead of use the classical calibration approach, it is advisable, from the point of view of minimizing the sum of squared errors of prediction, to use the inverse regression. A guide on how to get it right is in figure 6.

Even if the LR hypotheses with regressor the absorbance and the concentration as response are not accomplished, the approximation of the inverse regression remains valid: to carry out predictions it is not necessary for the hypotheses to be fulfilled since the inverse regression approach relies on OLS, which does not depend on any hypothesis. Moreover, the greater predictive power of the inverse regression, compared with that of classical calibration, gives support to its use. This fact is founded on the fact that inverse regression minimizes the sum of the squared error of the predictions for the concentration given the absorbance, but it is also shown empirically in this work by a toy example in §5 and two more examples, one with real data and the other built by simulation, in appendix B.

That in the classical calibration approach the LR hypotheses are fulfilled, is nothing more than an entelechy: how to be sure of the normality of the absorbance distribution given the concentration value, which is assumed to be fixed (and determined without error, despite the fact that measurement errors are unavoidable), and of the rest of the hypotheses? Despite the (possible but not usual) utilization of methods for studying the goodness of fit of the observations to them, the assumption of the hypotheses of a model is always a delicate subject that could be considered, in a sense, a matter of faith. Evaluating the predictive capacity of a model by means of the sum of the squares of the errors of prediction is not.

Even in the simulation example presented in appendix B, in which the absorbance values have been simulated from those of the concentration, that are fixed, according to the equation of a straight line with an additive Gaussian noise, that is, in such a say that it can be assumed that the LR hypotheses are fulfilled with the concentration as regressor and the absorbance as output variable (classical calibration), from a predictive point of view it turns out that the inverse regression approach surpasses

the classical calibration. In other words: leaving aside assumptions that could, or not, be accomplished (that in the case to be fulfilled allow to deduce some statistical properties for the linear regression model), if we are interested in prediction, the most appropriate would be to use the inverse regression approach.

It is true that in many applications the difference between the predicted concentrations obtained with both calibration curves is small, and therefore, for practical purposes, this error does not usually have great consequences. However, this does not justify overlooking the entanglement, which is important from a conceptual point of view. What is more, it could potentially have practical consequences, so it should be avoided. This paper aims to draw the attention of experimental scientists to this important issue and contribute to the eradication of this pitfall.

Data accessibility. All scripts used in this study are openly accessible through https://github.com/StochasticBiology/boolean-efflux.git. The data are provided in electronic supplementary material [20]. I have used simulated data that I have uploaded in a csv format file.
Competing interests. I declare I have no competing interests.
Funding. The author is supported by Ministerio de Ciencia, Innovación y Universidades, Gobierno de España, project ref. PGC2018-097848-B-I0.
Acknowledgements. The author wishes to thank the anonymous referees and Associate Editor for careful reading and helpful comments that resulted in an overall improvement of the paper.

# Appendix A. The linear regression model

In this section, we will see the formulae relative to the linear regression model, which is a model to describe the linear relationship between two quantitative variables, namely $X$, which is the input or regressor, and $Y$, which is the output or predicted variable. In each scenario, which of the two variables should play the role of $X$, and which of $Y$, depends on the objective: the variable that has to play the role of $Y$ is the one for which we want to obtain a prediction given a known value for the other variable (which, then, will play the role of $X$). This asymmetry between the variables is a factor to take into account, since it could be a source of confusion. Indeed, it is very important to resolve this issue at the beginning, before building the model, since making the wrong decision will lead, as has been explained above that is common in instrument calibration by spectrometry, to predictions subject to greater error, being precisely to highlight and clarify this matter, the motivation of this paper.

The linear regression model of $Y$ over $X$ is a straight line whose equation is the one that better fits the data, which is a set of $n > 2$ pairs of values of the variables $X$ and $Y$, say $(x_1, y_1)$, $(x_2, y_2)$, …, $(x_n, y_n)$, and is given by

$$y = b_0 + b_1 x,$$

where $b_0$ and $b_1$ are obtained from the data in this way

$$\left.\begin{array}{l} b_1 = \dfrac{S_{xy}}{S_{xx}}, \quad \text{with } \bar{x} = \dfrac{1}{n}\sum_{i=1}^{n} x_i, \ \bar{y} = \dfrac{1}{n}\sum_{i=1}^{n} y_i, \\[2ex] S_{xy} = \sum_{i=1}^{n} x_i y_i - n\,\bar{x}\,\bar{y}, \ S_{xx} = \sum_{i=1}^{n} x_i^2 - n\,(\bar{x}^2) \\[2ex] b_0 = \bar{y} - b_1\,\bar{x} \end{array}\right\} \quad \text{(A 1)}$$

(Note that the asymmetry between $X$ and $Y$ is reflected in the expressions to obtain the coefficients of the straight line $b_0$ and $b_1$.)

In what sense is the regression line the one that best approximates the data? In which it is the one that minimizes the sum of the squared errors, denoted by $e_i$, which are the difference between the observed value of the variable $Y$ when the variable $X$ takes the value $x_i$, which is $y_i$, and the prediction given by the regression straight line, which is $\hat{y}_i = b_0 + b_1 x_i$, that is, $e_i = y_i - \hat{y}_i$. If the relationship between $X$ and $Y$ were perfectly explained by the straight line (hypothetical and deterministic situation), then $e_i = 0$ for $i = 1, …, n$.

By imposing this criterion we can easily find (A 1). This is the well-known ordinary least squares (OLS) method, due to Carl F. Gauss. To apply this method, we must derive

$$\text{SSE} = \sum_{i=1}^{n} e_i^2 = \sum_{i=1}^{n}(y_i - \hat{y}_i)^2 = \sum_{i=1}^{n}\left(y_i - (b_0 + b_1 x_i)\right)^2,$$

with respect to $b_0$ and $b_1$, and set these two derivatives to zero. Indeed, we obtain

$$-2\sum_{i=1}^{n}\left(y_i - (b_0 + b_1\,x_i)\right) = 0 \quad \text{and} \quad -2\sum_{i=1}^{n}\left(y_i - (b_0 + b_1\,x_i)\right)x_i = 0.$$

From the first we get

$$\sum_{i=1}^{n}\left(y_i - (b_0 + b_1\,x_i)\right) = 0 \Leftrightarrow \sum_{i=1}^{n}y_i - n\,b_0 - b_1\sum_{i=1}^{n}x_i = 0$$

$$\Leftrightarrow \bar{y} - b_0 - b_1\,\bar{x} = 0 \Leftrightarrow b_0 = \bar{y} - b_1\,\bar{x},$$

and from the second, by substituting the expression obtained for $b_0$, we finally have that

$$\sum_{i=1}^{n}\left(y_i - (b_0 + b_1\,x_i)\right)x_i = 0 \Leftrightarrow \sum_{i=1}^{n}x_i\,y_i - b_0\sum_{i=1}^{n}x_i - b_1\sum_{i=1}^{n}x_i^2 = 0$$

$$\Leftrightarrow \sum_{i=1}^{n}x_i\,y_i - (\bar{y} - b_1\bar{x})n\bar{x} - b_1\sum_{i=1}^{n}x_i^2 = 0$$

$$\Leftrightarrow \sum_{i=1}^{n}x_i\,y_i - n\,\bar{x}\,\bar{y} + b_1\,n\,(\bar{x})^2 - b_1\sum_{i=1}^{n}x_i^2 = 0$$

$$\Leftrightarrow b_1 = \frac{\sum_{i=1}^{n}x_i\,y_i - n\,\bar{x}\,\bar{y}}{\sum_{i=1}^{n}x_i^2 - n\,(\bar{x})^2} = \frac{S_{xy}}{S_{xx}}$$

(further verification that it is indeed a minimum is necessary, although we will not go into details). A value that is used as a measure of how well the regression straight line approximates to the $n$ point, is the determination coefficient or $R$-squared, being defined by

$$R^2 = \frac{\left(\sum_{i=1}^{n}x_i\,y_i - n\,\bar{x}\,\bar{y}\right)^2}{\left(\sum_{i=1}^{n}x_i^2 - n\,(\bar{x})^2\right)\left(\sum_{i=1}^{n}y_i^2 - n\,(\bar{y})^2\right)} = \frac{S_{xy}^2}{S_{xx}\,S_{yy}},$$

with $S_{yy} = \sum_{i=1}^{n}y_i^2 - n\,(\bar{y})^2$, which is between 0 and 1 and is interpreted as the proportion of the total variability of the data that is explained by the regression straight line. The closer to 1 is $R$, the better the linear approximation of the relationship between variables $X$ and $Y$. Its square root, with the sign of the slope $b_1$, is the well-known Pearson's correlation coefficient $r \in [-1,\ 1]$.

## A.1. The hypotheses of the regression model (LR hypotheses)

The regression model assumes that for each fixed value of the variable $X$, $x_i$ ($i = 1, \ldots, n$), the random variable $Y$, which is denoted in this case by $Y_i$, has Gaussian distribution with a mean which is a linear function of $x_i$, say $\gamma_0 + \gamma_1\,x_i$, where $\gamma_0$ and $\gamma_1$ are parameters independent of $i$, and with variance $\sigma^2 > 0$, which is also a parameter independent of $i$, that is, we assume that

$$Y_i \sim N(\gamma_0 + \gamma_1\,x_i,\ \sigma^2) \quad i = 1, \ldots, n.$$

Moreover, we assume that the random variables $Y_1, \ldots, Y_n$ are independent. In other words,

$$Y_i = \gamma_0 + \gamma_1\,x_i + \delta_i \quad i = 1, \ldots, n, \tag{A 2}$$

where $\delta_1, \ldots, \delta_n$ are independent and identically distributed random variables, $N(0,\ \sigma^2)$. These are the LR hypotheses that are needed in order to perform statistical inferences. We assume them in the remainder of appendix A. In this context, $b_0$ and $b_1$, the coefficients of the regression straight line, are the estimations of the parameters of the model $\gamma_0$ and $\gamma_1$, respectively, obtained from data, that is, $\hat{\gamma}_0 = b_0$ and $\hat{\gamma}_1 = b_1$. The estimation of parameter $\sigma^2$ is

$$\frac{\sum_{i=1}^{n}e_i^2}{n-2} = \frac{\text{SSE}}{n-2} = \text{MSE}. \tag{A 3}$$

## A.2. The coefficient estimates

Consider the estimations $b_0$ and $b_1$ of the coefficients of the linear regression model (respectively, $\gamma_0$ and $\gamma_1$ in equation (A 2)) given by (A 1). If in (A1) we substitute the observations $y_i$ by the random variables from which they are assumed to be realizations, $Y_i$, we obtain the expressions in (A 4) of the estimators of the coefficients, say $B_0$ and $B_1$, which are random variables from which the estimations $b_0$ and $b_1$, respectively, are realizations.

$$\left.\begin{array}{l} B_1 = \dfrac{S_{xY}}{S_{xx}}, \quad \text{with } \bar{Y} = \dfrac{1}{n}\sum_{i=1}^{n} Y_i, \quad S_{xY} = \sum_{i=1}^{n} x_i\,Y_i - n\,\bar{x}\,\bar{Y} \\[2mm] B_0 = \bar{Y} - B_1\,\bar{x} \end{array}\right\} \tag{A 4}$$

The Gauss–Markov theorem[1] says that if the hypothesis of the linear regression model, LR hypotheses, are satisfied, the estimators $B_0$ and $B_1$ are unbiased, that is, their distributions are centred at the corresponding coefficients

$$E\,(B_1) = \gamma_1, \quad E(B_0) = \gamma_0,$$

($E$ denotes expectation of a random variable, that is, its mean value), and they are the tightest possible in the sense that they have the smallest variance among all possible estimators of the coefficients that are linear functions of the variables $Y_1, \ldots, Y_n$. Then, they are the best linear unbiased estimators (BLUE) of the coefficients of the linear regression model.

With regard to the other parameter of the model, $\sigma^2$, its estimation is given by (A 3), which is the realization of the estimator $\widehat{\sigma^2}$, a random variable independent of $B_0$ and $B_1$ defined by

$$\widehat{\sigma^2} = \frac{\sum_{i=1}^{n} E_i^2}{n-2}, \quad \text{with } E_i = Y_i - \left(B_0 + B_1\,x_i\right), \quad \text{that verifies } \frac{\widehat{\sigma^2}}{\sigma^2}\,(n-2) \sim \chi_{n-2}^2.$$

## A.3. The analysis of the variance (ANOVA) for the linear regression model

The principles and methodology of ANOVA (ANalysis Of the VAriance) can be applied to study if there is a linear relationship between two variables $X$ and $Y$. Specifically, we will carry on a statistical test for the hypotheses

$$\left.\begin{array}{l} H_0 : \gamma_1 = 0 \\ H_1 : \gamma_1 \neq 0 \end{array}\right\}$$

($H_0$ is the null statistical hypothesis that corresponds to 'no linear relationship between the variables', while the alternative $H_1$ is the opposite). Considering that quantities $x_1, \ldots, x_n$ are fixed, the total variability of the observations is measured by the 'total sum of squares' $\mathrm{SST} = \sum_{i=1}^{n}(y_i - \bar{y})^2$, which can be decomposed as

$$\mathrm{SST} = \sum_{i=1}^{n} \mathrm{e}_i^2 + b_1^2 \sum_{i=1}^{n}(x_i - \bar{x})^2 = \mathrm{SSE} + b_1^2\,S_{xx},$$

where SST has $n-1$ associated degrees of freedom (over the $n$ quantities $y_i - \bar{y}$, there is only one linear restriction: $\sum_{i=1}^{n}(y_i - \bar{y}) = 0$), SSE has $n-2$ degrees of freedom since we sum the squares of $n$ terms with two independent linear restrictions: $\sum_{i=1}^{n} e_i = 0$ and $\sum_{i=1}^{n} e_i\,(x_i - \bar{x}) = 0$, and finally $b_1^2\,S_{xx}$ has 1 degree of freedom since it is fixed.

The statistical test of hypotheses consists in rejecting $H_0$ if $f = b_1^2\,S_{xx}/\mathrm{MSE}$, with $\mathrm{MSE} = \mathrm{SSE}/(n-2)$, is 'big enough', that means greater than a tabulated value. As it can be seen (we do not give the details here) that $f$ is the realization of a random variable $F$ with distribution Fisher's $F$ with 1 and $n-2$ degrees of freedom if the null hypothesis $H_0$ is true, that is,

$$F = \frac{B_1^2\,S_{xx}}{\left(\sum_{i=1}^{n} E_i^2/(n-2)\right)} \sim F_{1,n-2} \quad \text{if } \gamma_1 = 0,$$

[1]As explained in [16], the method of OLS was developed by Gauss in *Theoria combinationis observationum erroribus minimis obnoxiae* (1823), where a first proof of an early version of the theorem is given. Markov rediscovered the same result and included it in his book *Wahrscheinlichkeitsrechnung* (1912), the year in which Fisher converts least squares into a general estimation method in statistics. The terminology Gauss–Markov theorem comes from Neyman. For historical details, see [17].

the hypothesis null $H_0$ is rejected with a significance level $\alpha$ (then, a linear relationship between the variables is accepted) if

$$p\text{-value} = P(F_{1,n-2} > f) < \alpha.$$

Calculations necessary to obtain $f$ are usually carried out with help of the ANOVA table:

Analysis of variance (ANOVA) table

| source of variation response $Y$ | degree of freedom (d.f.) | sum of squares (sum Sq) | mean square (mean Sq) | $F$-value |
|---|---|---|---|---|
| regressor $X$ | 1 | $b_1^2 S_{xx}$ | $b_1^2 S_{xx}$ | $f = b_1^2 S_{xx}/\text{MSE}$ |
| residuals (error) | $n-2$ | $\text{SSE} = \sum_{i=1}^{n} e_i^2$ | $\text{MSE} = \text{SSE}/(n-2)$ | |
| total | $n-1$ | $\text{SST} = \sum_{i=1}^{n}(y_i - \bar{y})^2$ | | |

## A.4. Predicting with the linear regression model

Given a value for the variable $X$, let us say $x_0$, the straight line equation is used to predict the corresponding for the variable $Y$, which is denoted by $\hat{y}|_{x_0}$, in the following way:

$$\hat{y}|_{x_0} = b_0 + b_1 x_0,$$

and it can be carried out as long as the value $x_0$ is found within the range of values given by $x_1, \ldots, x_n$, and if the linear approximation is good ($R^2$ big enough).

## A.5. Confidence intervals for the coefficients

Fixed $\gamma \in (0, 1)$ as confidence level, the confidence intervals for the coefficients of the regression straight line are

$$\gamma_1 : b_1 \pm t_{1-(\alpha/2)}^{n-2} \sqrt{\frac{\left(\sum_{i=1}^{n} e_i^2/(n-2)\right)}{S_{xx}}} \quad \text{and} \quad \gamma_0 : b_0 \pm t_{1-(\alpha/2)}^{n-2} \sqrt{\frac{\left(\sum_{i=1}^{n} e_i^2/(n-2)\right)\left(\sum_{i=1}^{n} x_i^2/n\right)}{S_{xx}}}$$

where $\alpha = 1 - \gamma$ and $t_{1-(\alpha/2)}^{n-2}$ is the critical value for the distribution Student's $t$ with $n-2$ degrees of freedom, $t_{n-2}$, such that the probability that this distribution gives a value greater than the critical value is $\alpha/2$ (that is, given $\omega \in (0, 1)$, $t_{\omega}^{n-2}$ denotes the real number such that $P(t_{n-2} < t_{\omega}^{n-2}) = \omega$).

## A.6. Confidence interval for the prediction

Fixed a value for the variable $X$, say $x_0$, and $\gamma \in (0, 1)$ as confidence level, the confidence interval for the prediction for the variable $Y$, $\hat{Y}|_{x_0} = \gamma_0 + \gamma_1 x_0$ (which can be thought as a new parameter, function of $\gamma_0$ and $\gamma_1$, whose estimation is $\hat{y}|_{x_0} = b_0 + b_1 x_0$) is

$$\hat{y}|_{x_0} \pm t_{1-(\alpha/2)}^{n-2} \sqrt{\left(\frac{\sum_{i=1}^{n} e_i^2}{n-2}\right)\left(\frac{1}{n} + \frac{(x_0 - \bar{x})^2}{S_{xx}}\right)}.$$

The value of $x_0$ that minimizes the length of the confidence interval for the prediction is $x_0 = \bar{x}$. As $x_0$ moves away from $\bar{x}$ (by excess or by default) the length increases symmetrically.

**Table 4.** Example of practical calibration (table 3 in [19]): five replications of the absorbance reading for any of the 14 fixed concentrations.

| concentration | absorbance | | | | |
|---|---|---|---|---|---|
| 0 | 0 | 0 | 0 | 0 | 0 |
| 1 | 0.053 | 0.053 | 0.054 | 0.054 | 0.055 |
| 2 | 0.092 | 0.092 | 0.092 | 0.092 | 0.092 |
| 3 | 0.130 | 0.134 | 0.129 | 0.129 | 0.128 |
| 4 | 0.181 | 0.181 | 0.181 | 0.179 | 0.180 |
| 5 | 0.209 | 0.208 | 0.208 | 0.207 | 0.207 |
| 6 | 0.265 | 0.265 | 0.264 | 0.262 | 0.264 |
| 7 | 0.324 | 0.324 | 0.324 | 0.324 | 0.324 |
| 8 | 0.354 | 0.352 | 0.352 | 0.352 | 0.354 |
| 9 | 0.381 | 0.379 | 0.381 | 0.379 | 0.381 |
| 10 | 0.430 | 0.430 | 0.430 | 0.430 | 0.430 |
| 20 | 0.881 | 0.880 | 0.880 | 0.880 | 0.882 |
| 40 | 1.576 | 1.575 | 1.576 | 1.576 | 1.576 |
| 60 | 2.062 | 2.062 | 2.062 | 2.060 | 2.062 |

## A.7. Statistical hypotheses testing

Fixed a significance level $\alpha \in (0,\ 1)$, the statistical test of hypotheses for the parameters of the regression model are:

$$\gamma_1: \text{statistic } t = \frac{b_1 - \gamma_1^0}{\sqrt{\left(\sum_{i=1}^n e_i^2/(n-2)\right)/S_{xx}}}, \quad \gamma_0: \text{statistic } t = \frac{b_0 - \gamma_0^0}{\sqrt{\left(\sum_{i=1}^n e_i^2/(n-2)\right)\left(\sum_{i=1}^n x_i^2/n\right)/S_{xx}}}$$

| alternative hypothesis$(\gamma_1^0,\ \gamma_0^0 \text{ fixed })$ | | accepted if | | $p$-value |
|---|---|---|---|---|
| $H_1: \gamma_1 > \gamma_1^0 \quad / \quad \gamma_0 > \gamma_0^0$ | $\longrightarrow$ | $t > t_{1-\alpha}^{n-2}$ | $\longrightarrow$ | $P(t_{n-2} > t)$ |
| $H_1: \gamma_1 < \gamma_1^0 \quad / \quad \gamma_0 < \gamma_0^0$ | $\longrightarrow$ | $t < t_{\alpha}^{n-2}$ | $\longrightarrow$ | $P(t_{n-2} < t)$ |
| $H_1: \gamma_1 \neq \gamma_1^0 \quad / \quad \gamma_0 \neq \gamma_0^0$ | $\longrightarrow$ | $\lvert t \rvert > t_{1-\alpha/2}^{n-2}$ | $\longrightarrow$ | $2\,P(t_{n-2} > \lvert t \rvert)$ |

## A.8. Prediction interval

Fixed a value for the variable $X$, say $x_0$, and $\gamma \in (0,\ 1)$ as confidence level, the prediction interval is an interval 'of the most probable values' for the variable $Y$, that when $X = x_0$ we denote by $Y_0$, that is, $Y_0 = \gamma_0 + \gamma_1 x_0 + \delta_0$ with $\delta_0 \sim N(0,\ \sigma^2)$ independent of $\delta_1, \ldots,\ \delta_n$. Informally speaking, the prediction interval is an interval where the variable $Y_0$ takes values with probability $\gamma$, and has the expression

$$\hat{y}\rvert_{x_0} \pm t_{1-(\alpha/2)}^{n-2} \sqrt{\left(\frac{\sum_{i=1}^n e_i^2}{n-2}\right)\left(1 + \frac{1}{n} + \frac{(x_0 - \bar{x})^2}{S_{xx}}\right)}. \tag{A 5}$$

## A.9. Prediction interval for classical calibration

The problem with classical calibration is that to make predictions we have to deal with the reciprocal of the slope, which follows a Gaussian distribution under the hypothesis of the linear regression model. The reciprocal of a Gaussian random variable has infinite variance (then, the mean squared error is infinite), but although an asymptotic approximation can be derived using the Delta method (see [14]), it has limitations. By formulae (4.32) and (4.32$a$) in [18, p. 169], for any absorbance $A_i$, the corresponding prediction interval using the classical calibration and under the hypothesis of the linear regression model,

**Table 5.** Predictions with the two methods: classical calibration and inverse regression, and corresponding radius of the prediction intervals and errors, for data in table 4. In italics the maximum $R^2$ and the minimum standard error s.e.

| $A_i$ | $c_i$ | predictions $\widehat{c_i}$ classical $\frac{A_i - \beta_0}{\beta_1}$ | inverse $\alpha_0 + \alpha_1 A_i$ | prediction interval radius classical (a) | inverse (b) | errors $c_i - \widehat{c_i}$ classical $e_i$ | inverse $\varepsilon_i$ |
|---|---|---|---|---|---|---|---|
| ⋮ | ⋮ | ⋮ | ⋮ | ⋮ | ⋮ | ⋮ | ⋮ |
| 0.181 | 4 | 3.62591020 | 3.7126938 | 2.582782 | 2.576339 | 0.374089799 | 0.28730621 |
| 0.179 | 4 | 3.56942588 | 3.6567619 | 2.582810 | 2.576365 | 0.430574117 | 0.34323815 |
| 0.180 | 4 | 3.59766804 | 3.6847278 | 2.582796 | 2.576352 | 0.402331958 | 0.31527218 |
| 0.209 | 5 | 4.41669065 | 4.4957409 | 2.582411 | 2.575986 | 0.583309349 | 0.50425915 |
| 0.208 | 5 | 4.38844849 | 4.4677749 | 2.582423 | 2.575998 | 0.611551508 | 0.53222512 |
| ⋮ | ⋮ | ⋮ | ⋮ | ⋮ | ⋮ | ⋮ | ⋮ |
| SSE = | | | | | | 187.5209 | 185.687 |
| MSE = SSE/(n − 2) = | | | | | | 2.75766 | 2.730692 |
| $se = \sqrt{MSE} =$, | | | | | | 1.66062 | *1.65248* |
| $R^2 = 1 - SSE/SST =$ | | | | | | 0.990124 | *0.9902206* |

is of the form $\widehat{c_i} \pm (a)$, where

$$(a) = t^{n-2}_{1-\frac{\alpha}{2}} \frac{1}{\beta_1} \sqrt{\left(\frac{\sum_{i=1}^n \widetilde{e}_i^2}{n-2}\right)\left(1 + \frac{1}{n} + \frac{(\widehat{c_i} - \bar{c})^2}{\sum_{i=1}^n c_i^2 - n\bar{c}^2}\right)}, \tag{A 6}$$

with $\widetilde{e}_i$ being the errors committed with the classical calibration, not to predict concentration from absorbance but to predict absorbance by concentration, that is, $\widetilde{e}_i = A_i - (\beta_0 + \beta_1 c_i)$. See also formula (5) in [14].

# Appendix B. Two more examples

## B.1. An example of practical calibration with real experimental data

The following example of practical calibration is borrowed from [19] and can be used to compare the approaches of classical calibration and inverse regression. The data (table 3 in [19]) are absorbance readings for potassium permanganate at 525 nm given by the scanning of the spectrophotometer for different concentrations. Specifically, a stock solution for standards was made by 0.072 g of potassium permanganate in 250.0 cm$^3$ standard flask, and standard working solutions are five replicates of each one, containing, respectively, 0, 1, 2, 3, 4, 5, 6, 7, 8, 9, 10, 20, 40, 60 mg dm$^{-3}$ of potassium permanganate, made by dilution of appropriate aliquots of the stock solution to 100.0 cm$^3$ with deionized water. Concentrations and measured absorbances are recorded in table 4 .

The two calibration curves given by (3.2) and (4.2) are

Classical calibration (curve of $A$ over $c$): $A = \beta_0 + \beta_1 c = 0.05261356 + 0.03540806 c$
Inverse regression (curve of $c$ over $A$): $c = \alpha_0 + \alpha_1 A = -1.349146 + 27.96597 A$.

For the difference between the absolute value of the errors in prediction with the classical calibration and the inverse regression (the former minus the latter) (table 5), we perform a one-sided Wilcoxon signed-rank test, which is the non-parametric counterpart of the $t$-test, to compare its median with 0 (the $p$-value of the Shapiro–Wilk test for normality is $1.976 \times 10^{-8}$***, meaning that we have enough evidence to reject the normality of the sample). The $p$-value of the one-sided Wilcoxon test with the alternative hypothesis: 'the median of the difference is greater than 0' is 0.0002126***; that shows a clear statistical significance in favour of inverse regression.

**Table 6.** First 10 observations of the simulated dataset. Note that original observation 8 has been deleted since the simulated absorbance for a concentration of 57 was the negative number −1.1404749.

| observation original order | $c_i$ | $A_i$ |
|---|---|---|
| 1 | 50 | 0.7376204 |
| 2 | 51 | 1.8321149 |
| 3 | 52 | 7.5390685 |
| 4 | 53 | 2.8829671 |
| 5 | 54 | 3.1188437 |
| 6 | 55 | 8.1835117 |
| 7 | 56 | 4.2675450 |
| 9 | 58 | 0.7379806 |
| 10 | 59 | 1.5506931 |
| 11 | 60 | 6.8808865 |
| ⋮ | ⋮ | ⋮ |

With respect to the prediction interval radius, for all the ($n = 70$) observations, the radius for the inverse regression is less than that of the classical calibration approach. We can perform a statistical test to check if the median of the difference of the prediction interval radius (classical calibration minus inverse regression) is significantly greater than 0. As the $p$-value for the Shapiro–Wilk of normality is $8.925 \times 10^{-14***}$, we reject normality and make the one-sided Wilcoxon signed-rank test, obtaining as $p$-value $1.793 \times 10^{-13***}$, that expresses a very high statistical significance in favour of the inverse regression approach.

The analysis of variance (ANOVA) table for regression (see appendix A) applied to the inverse regression approach is:

| source of variation response $c$ | d.f. | sum Sq | mean Sq | F-value |
|---|---|---|---|---|
| regressor $A$ | 1 | $\alpha_1^2 S_{AA} = 18801.81$ | $\alpha_1^2 S_{AA} = 18801.81$ | $f = 6885.344$ |
| residuals (error) | 68 | $SSE = \sum_{i=1}^n e_i^2 = 185.69$ | $MSE = 2.7307$ | |
| total | 69 | $SST = \sum_{i=1}^n (c_i - \bar{c})^2 = 18987.5$ | | |

with $S_{AA} = \sum_{i=1}^n (A_i - \bar{A})^2 = 24.04031$. The $p$-value for the statistical test with $H_0$: 'no linear relationship between $A$ and $c$', if the LR hypotheses can be reasonably assumed, is $P(F_{1,68} > 6885.344) < 2.2 \times 10^{-16***}$, and therefore we accept the linear relationship between concentration and absorbance.

## B.2. An example by simulation

Apart from the toy example in §5, and the practical calibration example with real experimental data in the first subsection of this appendix, now we will perform a simulation experiment consisting in the following. First, a dataset with some values of concentration and the corresponding absorbances have been created by simulation, in this way:

(i) Fix values for the concentration, $c_i$, from 50 to 500, with a step one by one: 50, 51, 52, ..., 499, 500 (a total of 451 values).

(ii) Compute the corresponding values of the absorbance $A_i$ by means of the linear expression with Gaussian additive noise (error),

$$A_i = 0.01 + 0.05\, c_i + \varepsilon_i, \quad i = 1, \dots, 451,$$

with $\varepsilon_i \sim N(\mu = 0, \sigma^2 = 10)$, all generated independently. We use the function rnom of R, and fix a random seed for reproducibility purpose with set.seed(123).

(iii) As it is possible that some values of the absorbance are negative, delete such observations. This will depend on the Gaussian values that have been randomly generated. In our case, we are left with a final number of $n = 447$.

**Table 7.** Sum of squared errors and mean sum of squared errors in the validation procedure for both approximations, classical calibration and inverse regression, with $k$-fold cross-validation, $k = 10$, and difference in the mean sum of squared errors between the approximations (classical calibration minus inverse regression).

| fold | sum of squared error | | mean sum of squared error | | |
|---|---|---|---|---|---|
| | classical | inverse | classical | inverse | difference |
| 1 | 268747.2 | 221536.15 | 6398.742 | 5274.670 | 1124.07218 |
| 2 | 207565.9 | 195265.34 | 4942.046 | 4649.175 | 292.87117 |
| 3 | 162817.2 | 137581.22 | 3876.600 | 3275.743 | 600.85696 |
| 4 | 148482.3 | 92484.90 | 3535.294 | 2202.022 | 1333.27201 |
| 5 | 198475.3 | 180907.94 | 4725.602 | 4307.332 | 418.26981 |
| 6 | 158211.0 | 105958.98 | 3766.930 | 2522.833 | 1244.09681 |
| 7 | 128881.1 | 112011.31 | 3068.597 | 2666.936 | 401.66138 |
| 8 | 128828.6 | 130050.20 | 3067.347 | 3096.433 | −29.08651 |
| 9 | 157612.6 | 84083.97 | 3752.680 | 2001.999 | 1750.68097 |
| 10 | 173684.0 | 147785.03 | 3544.572 | 3016.021 | 528.55118 |
| average: | | | 4067.841 | 3301.316 | 766.5246 |

**Table 8.** Average mean sum of squared errors for both approximations, classical calibration and inverse regression, with $k$-fold cross-validation, $k = 10$, $p$-value for the one-sided $t$-test to compare the differences in the mean/median, and $p$-value of the exact binomial test in favour of the inverse regression (except those marked with †, which are in favour of the classical calibration), with the number of folds, out of the 10 there are, for which the mean sum of squared errors is greater for the inverse regression than for the classical calibration in brackets. All the $p$-values are significant except those for $\sigma^2 < 2$.

| $\sigma^2$ | average mean sum of squared errors | | $p$-value | $p$-value |
|---|---|---|---|---|
| | classical | inverse | $t$-test/Wilcoxon test | exact binomial test |
| 0.01 | 4.014717 | 4.01434 | 0.4754 | (6) 0.2050781† |
| 0.1 | 40.17608 | 40.10235 | 0.3534 | (6) 0.2050781† |
| 0.5 | 201.1428 | 199.0806 | 0.1815 | (5) 0.2460938 |
| 1 | 402.6806 | 394.2868 | 0.1016 | (3) 0.117187500 |
| 2 | 806.484 | 772.9201 | 0.04209* | (2) 0.043945310* |
| 3 | 1211.606 | 1133.598 | 0.01526* | (1) 0.009765625** |
| 4 | 1617.850 | 1481.911 | 0.007936** | (1) 0.009765625** |
| 5 | 2009.188 | 1801.027 | 0.004883** | (2) 0.043945310* |
| 6 | 2439.265 | 2135.996 | 0.003818** | (1) 0.009765625** |
| 7 | 2832.297 | 2437.114 | 0.003476** | (1) 0.009765625** |
| 8 | 3243.029 | 2735.674 | 0.002206** | (1) 0.009765625** |
| 9 | 3654.897 | 3023.564 | 0.001446** | (1) 0.009765625** |
| 10 | 4067.841 | 3301.316 | 0.0009752*** | (1) 0.009765625** |
| 20 | 8539.498 | 5685.383 | 0.0003228*** | (1) 0.009765625** |
| 30 | 13335.26 | 7548.766 | 0.0009766*** | (0) 0.0009765625*** |

As usual, *, ** and *** denote significance at 5%, 1% and 0.1% levels, respectively.

The first 10 observations of the 447 have been recorded in table 6. For the dataset with $n = 447$ observations, we obtain 0.9020323 as Pearson's correlation coefficient, and 0.9060091 if instead we compute Spearman's correlation coefficient, both reflecting a good linear relationship between concentration values and the corresponding simulated absorbances.

Second, we use $k$-fold cross-validation with $k = 10$ to evaluate the prediction error with the two approaches, classical calibration and inverse regression. Indeed, we randomly order the $n$ instances (using the `sample` function of R), and then divide the observations into 10-folds, the first 9 composed of $\lfloor n/10 \rfloor$ observations (in this case, 44), and the last with the rest (51 observations). Then, for each fold:

(a) We reserve the fold for validation and learn the linear regression models (to follow the two approaches) with the rest of the folds as a learning (training) set.
(b) Once learned the two linear regression models, we follow the two approaches to predict, for each of the instances in the validation set, the concentration value from the known absorbance.
(c) As we know the observed concentration value corresponding to the absorbance of any observation in the validation set, we can compare the observed and the predicted values obtained with the two approaches.
(d) For any fold and approach, we compute the sum of the squared errors in making predictions and also divide by the number of instances minus 2, to compensate the fact that one of the folds has more observations than the other, obtaining in this way the mean sum of squared errors.

    Be careful: we are making predictions for the concentrations given the absorbances of new observations not seen by the regression models, which are the observations of the validation dataset. This is different from the usual situation in which we evaluate the predictive capacity of the model making predictions for the same observations that have been used to construct the model.
(e) Finally, we have two paired samples of size $k = 10$ of values of the mean sum of squared errors, that can be used to perform a statistical test to compare the two approaches from the point of view of their predictive power.

In table 7, we have recorded the mean sum of squared errors for each fold.

For the difference between the mean sum of squared errors with the classical calibration and the inverse regression (the former minus the latter), we can perform a one-sided $t$-test to compare its mean with 0 (since the Shapiro–Wilk test for normality gives a $p$-value of 0.5422, which implies that we do not have enough evidence to reject the normality of the sample). The $p$-value of the one-sided $t$-test with the alternative hypothesis: 'the mean of the difference is greater than 0' is 0.0009752[***], giving a very high statistical significance in favour of inverse regression being better than classical calibration (less mean sum of squared errors when predicting new cases). If instead, we had used the non-parametric Wilcoxon signed-rank test, not assuming normality of the sample of the differences, the one-sided $p$-value continues to be very small, 0.001953[**], showing statistical significance in the same sense.

Finally, it is also possible to compute the $p$-value of the exact binomial test in favour of the inverse regression, taking into account that out of 10 cases, there are nine in which the mean sum of squared errors is greater for the classical calibration, and only one in which it is less,

$$p\text{-value} = P(B(10,\ p = 0.5) = 1) = \binom{10}{1} 0.5^{10} = 10 \times 0.5^{10} = 0.009765625^{**}$$

(showing significance at 1% level).

As a conclusion, we can see that even in this example, in which the absorbance values have been simulated from those of the concentration to be able to reasonably assume the LR hypotheses with the classical calibration approach, favouring this approach, from the perspective of predictive power it is better to use the approximation of the inverse regression instead, in concordance with the conclusions in [13].

To evaluate the possible effect of the variance $\sigma^2$, that we have chosen to be 10 to simulate the absorbance values up to now, we repeat the procedure with other possible values ranging from 0.01 to 30. In table 8, we record for any $\sigma^2$, the values that had been computed before for the case $\sigma^2 = 10$: the average of the mean sum of squared errors (both, for the classical calibration and the inverse regression), the $p$-value of the one-sided $t$-test (or Wilcoxon signed-rank test, as appropriate) to compare the differences (mean/median of the classical calibration greater than that of inverse regression), and the $p$-value of the exact binomial test in favour of the inverse regression. We can observe clear evidence in favour of the inverse regression approach if $\sigma^2$ is big ($\sigma^2 \geq 2$), and no differences when $\sigma^2$ is small, which agrees with intuition.

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
