## [Peer Review File · Royal Society Open Science]

Review History

RSOS-211103.R0 (Original submission)

Review form: Reviewer 1

Is the manuscript scientifically sound in its present form?

Yes

Are the interpretations and conclusions justified by the results?

No

Is the language acceptable?

Yes

Do you have any ethical concerns with this paper?

Yes

Have you any concerns about statistical analyses in this paper?

Yes

Recommendation?

Reject

Comments to the Author(s)

ID: RSOS-211103

Title: Misuse of Beer-Lambert Law and other calibration curves

The author performed a mathematical study on the Beer-Lambert Law. What the author propose is to use the concentration as the ordinate and absorbance as the abscissa on the linear equation adjust by least squares. This approach is already used for multivariate inverse methods such as Multiple Linear Regression (MLR) and others.

Figure of merits is used to qualify the calibration model, so the results of classical and the proposed figure of merits must be compared and discussed. I suggest to see the following manuscript <https://doi.org/10.1021/cr400455s>.

Others tools used to evaluate the efficiency of a calibration curve, besides the figure of merit, are the Analysis of Variance (ANOVA) and the evaluation of the residue graphics. Both ANOVA and residue must be evaluated, compared and discussed in both classical and the proposed strategy. Using the explained variance as the author used to say that the proposed inverse strategy is correct over the classical one is too poor.

Therefore, I do not see enough novelty to suggest the publication of the present manuscript. In addition, the present examples are not enough to prove that the classical approach present significantly differences to the one proposed on the present manuscript.

More examples are necessary to proof it, especially more examples using real data acquired by accurate instruments such as chromatography, Atomic Absorption Spectroscopy, high resolution mass spectroscopy, and others.

Review form: Reviewer 2**Is the manuscript scientifically sound in its present form?**

Yes

Are the interpretations and conclusions justified by the results?

Yes

Is the language acceptable?

Yes

Do you have any ethical concerns with this paper?

No

Have you any concerns about statistical analyses in this paper?

No

Recommendation?

Accept with minor revision (please list in comments)

Comments to the Author(s)

See attached file (Appendix A).

Decision letter (RSOS-211103.R0)

Dear Dr Delgado:

Title: Misuse of Beer-Lambert Law and other calibration curves
Manuscript ID: RSOS-211103

The editor assigned to your manuscript has now received comments from reviewers. We would like you to revise your paper in accordance with the referee and Subject Editor suggestions which can be found below (not including confidential reports to the Editor). Please note this decision does not guarantee eventual acceptance.

Please submit your revised paper before 22-Oct-2021. Please note that the revision deadline will expire at 00.00am on this date. If we do not hear from you within this time then it will be assumed that the paper has been withdrawn. In exceptional circumstances, extensions may be possible if agreed with the Editorial Office in advance. We do not allow multiple rounds of revision so we urge you to make every effort to fully address all of the comments at this stage. If deemed necessary by the Editors, your manuscript will be sent back to one or more of the original reviewers for assessment. If the original reviewers are not available we may invite new reviewers.

Yours sincerely,
Dr Ellis Wilde
Publishing Editor, Journals

On behalf of the Subject Editor Professor Anthony Stace and the Associate Editor Dr Annette Trunschke.

RSC Associate Editor

Comments to the Author:

Two contradictory review reports were received on the manuscript. However, even the reviewer who recommends rejection of the manuscript confirms that the work is mathematically correct. He only criticises that other approaches and the existing literature were not sufficiently discussed and that perhaps too few examples were given.

Therefore, I recommend the acceptance of the manuscript after major revisions.

The author should respond in detail to the objections and suggestions for improvement and submit an improved version of the manuscript. I think it is important to point out to the user (e.g. chemist) the problem raised by the author.

There are a few points that the author could improve:

In chapter 3, the error regarding the calibration points is calculated. But that is not of interest. We are interested in the error with respect to new measurement data.

There is no mention of how this error behaves. At least a reference to relevant mathematical theory (BLUE) would be desirable.

Is the error one makes practically relevant?

The author's example in chapter 5 is illustrative. But I wonder if it reflects reality. Are the measurement errors in real calibrations really that big? It would be much more convincing if the author gave some examples with real experimental data and analysed them.

RSC Subject Editor

Comments to the Author:

(There are no comments.)

Reviewers' Comments to Author:

Reviewer: 1

Comments to the Author(s)

ID: RSOS-211103

Title: Misuse of Beer-Lambert Law and other calibration curves

The author performed a mathematical study on the Beer-Lambert Law. What the author propose is to use the concentration as the ordinate and absorbance as the abscissa on the linear equation adjust by least squares. This approach is already used for multivariate inverse methods such as Multiple Linear Regression (MLR) and others.

Figure of merits is used to qualify the calibration model, so the results of classical and the proposed figure of merits must be compared and discussed. I suggest to see the following manuscript <https://doi.org/10.1021/cr400455s>.

Others tools used to evaluate the efficiency of a calibration curve, besides the figure of merit, are the Analysis of Variance (ANOVA) and the evaluation of the residue graphics. Both ANOVA and residue must be evaluated, compared and discussed in both classical and the proposed strategy. Using the explained variance as the author used to say that the proposed inverse strategy is correct over the classical one is too poor.

Therefore, I do not see enough novelty to suggest the publication of the present manuscript. In addition, the present examples are not enough to prove that the classical approach present significantly differences to the one proposed on the present manuscript. More examples are necessary to proof it, especially more examples using real data acquired by accurate instruments such as chromatography, Atomic Absorption Spectroscopy, high resolution mass spectroscopy, and others.

Reviewer: 2

Comments to the Author(s)

See attached file

Author's Response to Decision Letter for (RSOS-211103.R0)

See Appendix B.

RSOS-211103.R1 (Revision)

Review form: Reviewer 1

Is the manuscript scientifically sound in its present form?

Yes

Are the interpretations and conclusions justified by the results?

Yes

Is the language acceptable?

Yes

Do you have any ethical concerns with this paper?

No

Have you any concerns about statistical analyses in this paper?

No

Recommendation?

Accept as is

Comments to the Author(s)

No more additional comments.

Now, I suggest the publication as it is.

Review form: Reviewer 2

Is the manuscript scientifically sound in its present form?

Yes

Are the interpretations and conclusions justified by the results?

Yes

Is the language acceptable?

Yes

Do you have any ethical concerns with this paper?

No

Have you any concerns about statistical analyses in this paper?

No

Recommendation?

Accept as is

Comments to the Author(s)

The paper is acceptable, but I just have one comment. I thin the author misunderstood my comment about "significant figures". As an Engineer, to me this does not signify the illustrations in the paper (Figure 1, Figure 2 etc), which is how the author has interpreted it. Instead, by "too many significant figures" I meant that there were too many numbers after the decimal place when dealing with real or simulated data. As the paper concerns errors, it is obvious that data and errors in data could not be known to the level of precision displayed. So as this is a misunderstanding, and the author has made a significant change in response to it, I want to give her the opportunity to reinstate any figures she believes will add to the scientific understanding of the work. It was not my intention that she would have to take them out. If she is happy with the paper as it is, that's fine by me.

Decision letter (RSOS-211103.R1)

Dear Dr Delgado:

Title: Misuse of Beer-Lambert Law and other calibration curves

Manuscript ID: RSOS-211103.R1

It is a pleasure to accept your manuscript in its current form for publication in Royal Society Open Science. The chemistry content of Royal Society Open Science is published in collaboration with the Royal Society of Chemistry.

Yours sincerely,
Dr Ellis Wilde
Publishing Editor, Journals

On behalf of the Subject Editor Professor Anthony Stace and the Associate Editor Dr Annette Trunschke.

RSC Associate Editor
Comments to the Author:
(There are no comments.)

RSC Subject Editor
Comments to the Author:
The author has addressed the reviewers' criticism in detail and submitted a significantly improved version of the manuscript. Nevertheless, I would be grateful for the opinion of the original reviewers.

Reviewer(s)' Comments to Author:
Reviewer: 1
Comments to the Author(s)
No more additional comments.
Now, I suggest the publication as it is.

Reviewer: 2
Comments to the Author(s)
The paper is acceptable, but I just have one comment. I think the author misunderstood my comment about "significant figures". As an Engineer, to me this does not signify the illustrations in the paper (Figure 1, Figure 2 etc), which is how the author has interpreted it. Instead, by "too many significant figures" I meant that there were too many numbers after the decimal place when dealing with real or simulated data. As the paper concerns errors, it is obvious that data and errors in data could not be known to the level of precision displayed. So as this is a misunderstanding, and the author has made a significant change in response to it, I want to give her the opportunity to reinstate any figures she believes will add to the scientific understanding of the work. It was not my intention that she would have to take them out. If she is happy with the paper as it is, that's fine by me.

Appendix A

Misuse of Beer-Lambert Law and other calibration curves

R Delgado

The author has spotted a fundamental error in the use of many calibration curves which can be applied equally to the Beer-Lambert Law and to many other calibration curves for experimental analytical data. She neatly exposes this in the literature via examples, and we probably have all made this error from time to time. She then goes on to provide an easy fix. It is all related to the way round in which one plots one's graphs, deciding which is the dependent variable, and therefore from which variable the errors are calculated using the least squares method.

I found the paper clear and easy to follow. The basic idea is breathtakingly simple, important and useful. Although the paper is clearly mathematically correct, the author has helped to convince others (including me) by the use of an example. It shows that there are still things to learn and subtleties within the most basic of experimental procedures. In my view the paper could be published almost as it is, however I believe its clarity and impact will benefit from some minor changes, listed below.

Comments

1. On page 3, point (c), I think I get what the author intended to say, but the English must have lost something in the translation? By the way I do agree with the points she makes here.
2. The author has nicely taken a number of examples from the literature to evidence that this is a widespread and ongoing issue across multiple disciplines. However, she has not placed her contribution to this problem into the broader context of work in statistical analysis. Some references here would also be helpful. They may need to come from textbooks as the status quo is accepted normal practice. Is this paper a novel contribution to the field? Even if not, it is certainly timely and needed.
3. Page 7 typo: absorvance should be absorbance
4. Page 8. It would be very helpful to state how the example numbers were chosen. Are they a particular set that happen to enhance the scale of the problem? Were they generated from an original linear relationship with the addition of random errors (and if so were they generated using a Gaussian number generator?) Are the relationships for A over c and c over A on p8 the original curves or the calculated curves? If the calculated curves, why do they come before the data?
5. In section 5, I find the choice of the number of significant figures to present is not consistent (and sometimes seems rather too many)
6. The axes and axis labels of Fig 6 were too thin and small, respectively
7. This is a preference: I would have liked to see the estimated error on the gradient and intercept of the curves rather than R^2 . We know that the example is one for which we expect a linear relationship, so I don't find the use of R^2 particularly useful, however the errors in gradient and intercept would add to my understanding. (For quick access to the formulae see G Squires, Experimental Physics, CUP).
8. In the discussion of Table 2, it would be helpful to estimate the error in se for this dataset, assuming errors with a Gaussian distribution (again see Squires). From my reading the difference in se between the two approaches is likely to be quite a bit smaller than the error in the estimate of se . It would also be interesting to know whether this is always the case (though not a necessary change for this paper to be published).

Appendix B

Dear RSC Associate Editor of the journal
Royal Society Open Science

This is the cover letter for the resubmission of a major revised version of the paper entitled

Misuse of Beer-Lambert Law and other calibration curves

Manuscript Number: **RSOS-211103**

authored by Rosario Delgado,

for it to be considered for publication in this journal. Below we explain in detail how the paper has been changed in reply to the Associate Editor and the reviewers' comments. I attach the revised version with the main changes in red to facilitate the reading.

• General comments:

The paper have been revised following the indications of the Associate Editor and of two reviewers. In general terms:

- a) The first paragraph of the Introduction has been completed, and the last one (before “The organization fo the rest of the paper...”) has been rewritten. Conclusion (Section 6) has also been rewritten.
- b) Figures have been reduced (see the answer to item 5 of Reviewer #2 for details).
- c) Sections 2 and 3 are basically the same as before, except that I introduce the notations: *classical calibration* and *inverse regression* to denote the claimed wrong and correct approaches, respectively.
- d) Various paragraphs have been added at the end of Section 4, to place the contribution of the paper in context with the help of some new references. The paper is a novel approach to this field, although not strictly a novel contribution, and is firmly committed to *inverse regression*, giving different arguments in his favor, both from a methodological and a practical point of view.
- e) Major changes have been made to Section 5, from Table 2 onwards, to make the analysis carried out on the toy example more powerful and to better extract the statistical significances from data.
- f) Some subsections have been added to Appendix A (just “Appendix” in the previous version): “The coefficient estimates” (explaining the Gauss-Markov theorem and the BLUE), “The ANalysis Of the VAriance (ANOVA) for the linear regression model”, “Prediction interval” and “Prediction interval for classical calibration”.
- g) Appendix B with two new examples has been added: one is a practical calibration with real experimental data borrowed from paper [1]. The other is an example built by simulation.

• **Answers to RSC Associate Editor:**

Two contradictory review reports were received on the manuscript. However, even the reviewer who recommends rejection of the manuscript confirms that the work is mathematically correct. He only criticizes that other approaches and the existing literature were not sufficiently discussed and that perhaps too few examples were given.

Therefore, I recommend the acceptance of the manuscript after major revisions. The author should respond in detail to the objections and suggestions for improvement and submit an improved version of the manuscript. I think it is important to point out to the user (e.g. chemist) the problem raised by the author. There are a few points that the author could improve:

1. In chapter 3, the error regarding the calibration points is calculated. But that is not of interest. We are interested in the error with respect to new measurement data. There is no mention of how this error behaves.

Answer: Classically, when working with the linear regression methodology, the prediction errors of the observations that have been used to learn the model are considered. Regression goodness-of-fit measures are typically defined from these errors (standard error, R-squared (R^2), ...). But I agree that it is more interesting to know how well the model predicts new observations. Therefore, in the example in Appendix B constructed by simulation, the technique of k -fold cross-validation has been used to estimate the prediction errors with new observations not seen by the model.

2. At least a reference to relevant mathematical theory (BLUE) would be desirable. Is the error one makes practically relevant?

Answer: A subsection devoted to the estimation of the coefficients of the linear regression models has been added to the Appendix A, “The coefficients estimates”, where the Gauss-Markov theorem and the BLUE are introduced succinctly. Of course, that only applies to the coefficients in the *inverse regression* approach, provided the usual assumptions are met. It cannot, however, be applied to the *classical calibration* approach. And the same happens with the ANOVA for the linear regression model. Please, see the answer to the item 7 of Reviewer # 2 for details.

What is experimentally showed in the paper, with different examples, is that the prediction error made with the *inverse regression* is significantly lower than that with the *classical calibration*, regardless of whether the hypotheses of the linear regression model are fulfilled or not.

3. The author’s example in chapter 5 is illustrative. But I wonder if it reflects reality. Are the measurement errors in real calibrations really that big? It would be much more convincing if the author gave some examples with real experimental data and analyzed them.

Answer: Two new examples have been introduced and analyzed in the new Appendix B, in addition to the toy example in Section 5: the first one is an example of practical calibration with real experimental data from paper [1]; the second is an example constructed by using simulation with which the errors in predicting new observations have been estimated through a k -fold cross-validation procedure.

• **Answers to Reviewer # 1:**

The author performed a mathematical study on the Beer-Lambert Law. What the author propose is to use the concentration as the ordinate and absorbance as the abscissa on the linear equation adjust by least squares. This approach is already used for multivariate inverse methods such as Multiple Linear Regression (MLR) and others. Figure of merits is used to qualify the calibration model, so the results of classical and the proposed figure of merits must be compared and discussed. I suggest to see the following manuscript <https://doi.org/10.1021/cr400455s>. Others tools used to evaluate the efficiency of a calibration curve, besides the figure of merit, are the Analysis of Variance (ANOVA) and the evaluation of the residue graphics.

1. Both ANOVA and residue must be evaluated, compared and discussed in both classical and the proposed strategy. Using the explained variance as the author used to say that the proposed inverse strategy is correct over the classical one is too poor. Therefore, I do not see enough novelty to suggest the publication of the present manuscript.

Answer: The ANOVA analysis for the regression linear model has been added in the toy example (Section 5) and in the first example in Appendix B (and subsection “The ANalysis Of the VAriance (ANOVA) for the linear regression model” has been added to Appendix A). But this analysis only applies to the *inverse regression* approach, provided the usual assumptions are met, and not to the *classical calibration* approach, which invalidates it as a methodology to compare both approaches. Please, see the answer to the item 7 of Reviewer # 2 for details.

In the paper it is experimentally showed through three different examples, and I believe that with consistent and well-founded statistical methods, that regardless of whether the hypotheses of the linear regression model are fulfilled or not, the prediction error made following the *inverse regression* approach is significantly lower than that committed with the *classical calibration*, what would advise the use of the first.

2. In addition, the present examples are not enough to prove that the classical approach present significantly differences to the one proposed on the present manuscript. More examples are necessary to proof it, especially more examples using real data acquired by accurate instruments such as chromatography, Atomic Absorption Spectroscopy, high resolution mass spectroscopy, and others.

Answer:

In accordance with this criticism, two new examples have been added in the new Appendix B, in addition to the toy example in Section 5: an example of practical calibration with real experimental data borrowed from paper [1], and other example constructed by using simulation with which the errors in predicting new observations have been estimated through a k -fold cross-validation procedure, and both have been carefully analyzed.

• **Answers to Reviewer # 2:**

The author has spotted a fundamental error in the use of many calibration curves which can be applied equally to the Beer-Lambert Law and to many other calibration curves for experimental analytical data. She neatly exposes this in the literature via examples, and we probably have all made this error from time to time. She then goes on to provide an easy fix. It is all related to the way round in which one plots one's graphs, deciding which is the dependent variable, and therefore from which variable the errors are calculated using the least squares method. I found the paper clear and easy to follow. The basic idea is breathtakingly simple, important and useful. Although the paper is clearly mathematically correct, the author has helped to convince others (including me) by the use of an example. It shows that there are still things to learn and subtleties within the most basic of experimental procedures. In my view the paper could be published almost as it is, however I believe its clarity and impact will benefit from some minor changes, listed below.

1. On page 3, point (c), I think I get what the author intended to say, but the English must have lost something in the translation? By the way I do agree with the points she makes here.

Answer: I have changed the wording of the last part of the introduction to make it more understandable, following the referee's indication.

2. The author has nicely taken a number of examples from the literature to evidence that this is a widespread and ongoing issue across multiple disciplines. However, she has not placed her contribution to this problem into the broader context of work in statistical analysis. Some references here would also be helpful. They may need to come from textbooks as the status quo is accepted normal practice. Is this paper a novel contribution to the field? Even if not, it is certainly timely and needed.

Answer: I have added the last part of the current Section 4, to place the contribution of the paper in context with the help of new references [9], [10] and [14]. The paper is not strictly a novel contribution to the field, but it is a novel approach who is firmly committed to *inverse regression*, and gives different arguments in his favor, both from a methodological and a practical point of view.

3. Page 7 typo: absorvance should be absorbance

Answer: Fixed up.

4. Page 8. It would be very helpful to state how the example numbers were chosen. Are they a particular set that happen to enhance the scale of the problem? Were they generated from an original linear relationship with the addition of random errors (and if so were they generated using a Gaussian number generator?) Are the relationships for A over c and c over A on p8 the original curves or the calculated curves? If the calculated curves, why do they come before the data?

Answer: In this toy example the numbers were made up. Well, in fact they have been "cooked" from those that appear in Table 1 in the paper

M.N. Abdulla, M.A. M-Ali, R.T. Kadhim, 2012. Preparation and Analytical Study of New Chelating Resin Containing Tetracycline Drug. Journal of Basrah Researches (Sciences), vol. 38 (3.A)

<https://www.researchgate.net/publication/>

310479772_Preparation_and_Analytical_Study_of_New_Chelating_Resin_Containing_Tetracycline_

The reason is that the original data did not show significant differences between the two approximations, and that is why I modified them so that they did show differences, since it was a toy example to illustrate what is explained in the paper.

Being aware of the weakness of the paper presenting only this “cooked” example, I have added two more examples in Appendix B, one with real data borrowed by new reference [1], and the other with simulated data from an exact linear relationship ($A = 0.01 + 0.05 c$) adding random additive Gaussian errors. Even in this last situation, in which the absorbance values have been simulated from those of the concentration in such a way the hypothesis of the linear regression model are fulfilled, situation that would favor the *classical calibration* approach over the *inverse regression*, we see that the latter behaves better from a predictive point of view.

As the calibration curves on p8 (original version of the paper) were the calculated ones, I have repositioned them by exchanging their position with the data. I agree with the referee that this arrangement is more natural (p9 in the revised version).

5. In section 5, I find the choice of the number of significant figures to present is not consistent (and sometimes seems rather too many)

Answer: I have reduced the number of plots in Figures 1 to 3 from 3 to 2 in each one of them. In addition, I have merged old Figures 4 and 5 into new Figure 4, containing only two plots.

6. The axes and axis labels of Fig 6 were too thin and small, respectively

Answer: Fixed up.

7. This is a preference: I would have liked to see the estimated error on the gradient and intercept of the curves rather than R2. We know that the example is one for which we expect a linear relationship, so I do not find the use of R2 particularly useful, however the errors in gradient and intercept would add to my understanding. (For quick access to the formulae see G Squires, Experimental Physics, CUP).

Answer: I fully understand the referee’s comment, but there is a problem with it. Everything that falls within the statistical inference relative to the coefficients of the straight line, or to the error, is based on the hypotheses of the linear regression model (I have called them **LR hypotheses** in the revised version of the paper). Usually, in the *classical calibration* approach, it is assumed that these hypotheses are met (although usually no goodness-of-fit test is performed, so it is not an assumption made rigorously) with the straight line $A = \beta_0 + \beta_1 c$, but when we isolate the concentration from the absorbance, the obtained calibration curve (see formula (3.4)) is $c = b + m A$ with $b = -\beta_0/\beta_1$ and $m = 1/\beta_1$, and the hypotheses, if it was true that they were fulfilled, now they no longer have to do so. Indeed, in the new subsection “Prediction interval for classical calibration”, Appendix A, I explain that with this procedure, things stop working as expected: the reciprocal of a Gaussian random variable (β_1) has infinite variance and then the mean squared error is infinite, although an asymptotic approximation can be derived by using the Delta method. In this way, the prediction interval radius with the *classical calibration* approach is only an approximation, given by formula (6.6).

And what about the other approximation? If in this case the **LR hypotheses** could be assumed with the straight line $c = \alpha_0 + \alpha_1 A$, then there would be no problem to make statistical inferences, but it might easily not be possible to make this assumption. Nevertheless, if it would be possible to design the experiment to collect data in such a way that these hypotheses are reasonably fulfilled, things will work out fine.

For this reasons, since they do not depend on the hypotheses of the linear model, and because from the point of view of real applications this is what matters, I have focused on the issue of **prediction errors** to compare the two approximations, *classical calibration* and *inverse regression*.

8. In the discussion of Table 2, it would be helpful to estimate the error in se for this dataset, assuming errors with a Gaussian distribution (again see Squires). From my reading the difference in se between the two approaches is likely to be quite a bit smaller than the error in the estimate of se . It would also be interesting to know whether this is always the case (though not a necessary change for this paper to be published).

Answer: The same response as to the previous referee's comment, applies, since the *standard error* se is computed for the *classical calibration* approach from the prediction errors in estimate the concentration from the absorbance, and it is not possible to assume that the errors be normally distributed. I have added to Table 2 the difference of the absolute values of the prediction errors, and performed a statistical test of hypotheses to see if this difference is statistically significantly positive in average, which is the case, although not in a very forceful way (p-value between 0.05 and 0.10), indicating that the prediction errors (in absolute value) are greater in average with the *classical calibration* than with the *inverse regression*.

In addition, Table 3 has been added containing the *prediction interval radius* for the toy example, obtained approximately by using the Delta method for the *classical calibration* approach, and again arrive to the conclusion that the *inverse regression* approach is better than the *classical regression*, in this case in the sense that in average, the radius of the prediction intervals are greater for this second one.